# Imageable AuNP-ECM Hydrogel Tissue Implants for Regenerative Medicine

**DOI:** 10.3390/pharmaceutics15041298

**Published:** 2023-04-20

**Authors:** Malka Shilo, Ester-Sapir Baruch, Lior Wertheim, Hadas Oved, Assaf Shapira, Tal Dvir

**Affiliations:** 1The Shmunis School of Biomedicine and Cancer Research, Faculty of Life Sciences, Tel Aviv University, Tel Aviv 6997801, Israel; 2Department of Materials Science and Engineering, Faculty of Engineering, Tel Aviv University, Tel Aviv 6997801, Israel; 3The Center for Nanoscience and Nanotechnology, Tel Aviv University, Tel Aviv 6997801, Israel; 4Department of Biomedical Engineering, Faculty of Engineering, Tel Aviv University, Tel Aviv 6997801, Israel; 5Sagol Center for Regenerative Biotechnology, Tel Aviv University, Tel Aviv 6997801, Israel

**Keywords:** tissue engineering, myocardial infarction, gold nanoparticles, ECM-based hydrogel, MRI, gadolinium

## Abstract

In myocardial infarction, a blockage in one of the coronary arteries leads to ischemic conditions in the left ventricle of the myocardium and, therefore, to significant death of contractile cardiac cells. This process leads to the formation of scar tissue, which reduces heart functionality. Cardiac tissue engineering is an interdisciplinary technology that treats the injured myocardium and improves its functionality. However, in many cases, mainly when employing injectable hydrogels, the treatment may be partial because it does not fully cover the diseased area and, therefore, may not be effective and even cause conduction disorders. Here, we report a hybrid nanocomposite material composed of gold nanoparticles and an extracellular matrix-based hydrogel. Such a hybrid hydrogel could support cardiac cell growth and promote cardiac tissue assembly. After injection of the hybrid material into the diseased area of the heart, it could be efficiently imaged by magnetic resonance imaging (MRI). Furthermore, as the scar tissue could also be detected by MRI, a distinction between the diseased area and the treatment could be made, providing information about the ability of the hydrogel to cover the scar. We envision that such a nanocomposite hydrogel may improve the accuracy of tissue engineering treatment.

## 1. Introduction

Cardiovascular diseases are a major cause of morbidity and mortality in developing countries [1,2], and myocardial infarction (MI) captures a significant fracture of these diseases. In MI, a blockage in one of the coronary arteries leads to ischemic conditions in the left ventricle (LV) of the myocardium and, therefore, to significant death of contractile cardiac cells. Since cardiac muscle cells cannot proliferate, this process eventually leads to the formation of scar tissue, which reduces heart functionality [3,4]. The main solution today for patients with severe heart failure is heart transplantation. However, since heart donors are scarce, there is an urgent need to develop new strategies to promote heart regeneration [5,6,7,8].

In cardiac tissue engineering, functional cardiac patches are composed of the contracting cells of the heart and extracellular matrix (ECM)-mimicking materials. The ECM provides structural and functional support to the injected cardiac cells, allowing cell reorganization into a functional heart patch and maintaining their viability after transplantation. Such cardiac cell-containing hydrogels were injected directly into the diseased area of the heart to repopulate the injured myocardium and restore its function [9,10,11,12,13,14]. However, injectable cardiac cell-containing hydrogels often cannot be accurately introduced into the entire diseased volume of the myocardium. In many cases, mainly when employing injectable systems or transplantable cardiac patches, the treatment may be partial because it does not fully cover the diseased area. Therefore, the treatment may not be effective or even induce conduction disorders. As of today, there is no efficient imaging technique that enables live tracking of the injected treatment, ensuring full coverage of the diseased area.

Over the past years, several studies have demonstrated the ability of conductive nanostructures, particularly gold nanoparticles (AuNPs), to improve the properties of the engineered tissue [15,16]. We previously showed that ECM hydrogels supplemented with AuNPs enhance the electrical signal between cardiac cells, allowing the formation of functional cardiac tissue [13]. Additionally, the AuNPs were also used to absorb reactive oxygen species after ischemia and reperfusion of the heart. We showed that this combined treatment can regenerate the infarcted heart [13].

Here, we demonstrate, for the first time, the ability of the injectable nanocomposite hydrogels to accumulate within the diseased area of the heart after myocardial infarction, allowing to accurately image the treatment in relation to the actual scar tissue. Several studies have shown the use of functionalized AuNPs with gadolinium chelates to serve as a promising theranostic agent for magnetic resonance imaging (MRI) [17,18,19]. However, this strategy has not yet been investigated to image the transplanted engineered tissue in the infracted area over time. Such capability would provide a means to understand where the diseased area is, and how effective the treatment is, allowing to noninvasively assess by MRI the coverage of the diseased area by the injected composite hydrogel. This would allow physicians to track the treatment’s location in real time and, accordingly, allow to inject another batch of the composite hydrogel into areas it may have failed to reach. This may provide comprehensive treatment to the entire diseased area, providing a more efficient treatment (Figure 1).

## 2. Materials and Methods

### 2.1. AuNP Synthesis

Spherical AuNPs were synthesized using sodium citrate according to the methodology described by Enüstun and Turkevich [20]. Briefly, 41.4 µL of 50% (*w*/*v*) HAuCl_4_ (Sigma-Aldrich, St. Louis, MO, USA) was mixed with 20 mL of purified water and heated to boiling. Next, 404 µL of trisodium citrate (Merck, Darmstadt, Germany) was added under stirring. A few minutes later, the color of the solution changed from clear yellow to deep red. After the solution was cooled to room temperature, it was centrifuged at 12,000 RPM for 20 min to precipitate the nanoparticles.

### 2.2. Gadolinium-Coated Gold Nanoparticle (Gd-AuNP) Synthesis

Au-GdL NPs were prepared according to the method described by Ji-Ae Park et al. [21] with slight modification. The ligand (L), a conjugate of DTPA with cysteine, was prepared via the reaction of DTPA-bis(anhydride) with two equivalents of cysteine in DMF at 80 °C for 6 h. The subsequent reaction of L with Gd_2_O_3_ in water for 6 h under reflux led to the formation of GdL as a white solid. Coating of AuNPs with GdL was accomplished through the direct addition of an aqueous solution of GdL to a solution of AuNPs [22]. Stirring was continued for 4 h before Au-GdL nanoparticles were collected by centrifugation and successively washed with water and acetone. The final Gd-AuNP concentration was 0.23 M. Gd-AuNPs were characterized using TEM and FTIR.

### 2.3. AuNP and Gd-AuNP Characterization

The size, shape, and uniformity of AuNPs and Gd-AuNPs were measured using transmission electron microscopy (TEM) (JEM-1400 Plus, JEOL, Pleasanton, CA, USA). Samples were prepared by dropping 5 μL of NP solution on a copper grid and then left to dry overnight before imaging. Contrast images were acquired in TEM and captured using SIS Megaview III and iTEM (Olympus, Shinjuku, Tokyo, Japan). For Gd-AuNPs, negative staining using uranyl acetate was used for AuNP and Gd-AuNP samples. The absorption spectrum of AuNPs and Gd-AuNPs was evaluated using an ultraviolet–visible (UV/Vis) spectrophotometer (NanoDrop, Wilmington, DE, USA). The size and charge of AuNPs and Gd-AuNPs were tested using dynamic light scattering (DLS), and their zeta potential was determined (Zetasizer, Malvern, Worcestershire, UK). Furthermore, Fourier-transform infrared spectroscopy (FTIR) (Nicolet iS10 Mid Infrared FT-IR Spectrometer, Thermo Scientific, Waltham, MA, USA) analysis was performed for all synthesis steps: AuNPs, DTPA complex, GdL complex, and Gd-AuNPs.

### 2.4. ECM-Based Hydrogel Preparation

Omenta were decellularized as previously described [23]. Briefly, omenta from healthy pigs (Kibbutz Lahav) were washed with phosphate-buffered saline (PBS). Then, the tissues were transferred to a hypotonic buffer (10 × 10^−3^ M Tris, 5 × 10^−3^ M ethylenediaminetetraacetic acid (EDTA) and 1 × 10^−6^ M phenylmethanesulfonyl-fluoride, pH 8.0) for 1 h. Next, tissues were frozen and thawed three times in the hypotonic buffer. After that, tissues were washed gradually with 70% (*v*/*v*) ethanol and 100% ethanol for 30 min each. Next, lipids were extracted using three washes of 100% acetone, 30 min each, followed by a 24 h incubation in a 60/40 (*v*/*v*) hexane/acetone solution (solution was changed three times). Then, the tissue was washed in 100% ethanol for 30 min and incubated overnight at 4 °C in 70% ethanol. Next, the tissue was washed with PBS (pH 7.4) four times and incubated in 0.25% Trypsin-EDTA solution (Biological Industries, Kibbutz Beit-Haemek, Israel) overnight. Then, the tissue was washed with PBS and incubated in 1.5 M NaCl for 24 h (solution was changed three times), followed by washing in 50 × 10^−3^ M Tris (pH 8.0) and 1% Triton X-100 (Sigma-Aldrich) solution for 1 h. The tissue was washed in PBS followed by double-distilled water, and then frozen (−20 °C) and lyophilized. The dry, decellularized omentum was ground into powder (Wiley Mini-Mill, Thomas Scientific, Swedesboro, NJ, USA). The omentum was then enzymatically digested under stirring for 96 h at room temperature, in a 1 mg·mL^−1^ solution of pepsin (Sigma-Aldrich, 4000 U·mg^−1^) in 0.1 M HCl. Next, the pH of the decellularized omentum was adjusted to 7.4 using DMEM/F12 × 10 (Biological Industries) and 5 M NaOH, to reach final concentration of 1% (*w*/*v*).

#### 2.4.1. High-Resolution Scanning Electron Microscopy (HRSEM)

Samples were mounted onto aluminum stubs with conductive paint. High-resolution images were obtained using an HRSEM ZEISS Gemini 300.

#### 2.4.2. Energy-Dispersive X-ray Spectroscopy (EDX)

Gd-AuNP implants were imaged after mounting onto aluminum stubs with conductive paint and without additional coating using a X-Flash 6/60 Bruker HRSEM with a field-emission gun (FEG) electron source. Imaging was carried out under low vacuum with a high voltage of 20 kV and a working distance of 7.3 mm.

#### 2.4.3. Rheological Properties

Rheological measurements (n = 6) were performed using a Discovery HR-3 hybrid Rheometer (TA Instruments, New Castle, DE, USA) with 8 mm diameter parallel plate geometry and a Peltier plate to maintain the sample temperature. Samples were prepared by encapsulation of AuNPs or Gd-AuNPs, to reach a final hydrogel percentage of 0.2%. Samples were loaded at a temperature of 4 °C, which was then raised to 37 °C, to examine the solidification process. The oscillatory moduli of samples were tested at a fixed frequency of 0.8 rad·s^−1^ and a strain of 1%.

### 2.5. Induced Pluripotent Stem Cell (iPSC) Culture

iPSCs were generated from omental stromal cells and were a kind gift from Dr. Rivka Ofir from Ben Gurion University. The undifferentiated cells were cultivated on 10 cm culture plates precoated with Matrigel™ (Corning, NY, USA) diluted to 250 µg·mL^−1^ in DMEM/F12 (Biological Industries, Kibbutz Beit-Haemek, Israel). Cells were maintained in NutriStem™ (Biological Industries) medium containing 0.1% penicillin/streptomycin (Biological Industries) and cultured under a humidified atmosphere at 37 °C with 5% CO_2_. The medium was refreshed daily, and cells were passaged at 70% confluence by treatment with 1 mL of ReLeSR™ (Stemcell Technologies, Vancouver, BC, Canada).

#### 2.5.1. Cardiomyocyte Differentiation from iPSCs

Prior to differentiation, cells were dissociated with Accutase™ (StemCell Technologies, Vancouver, BC, Canada) and passaged to six-well plates. Until 100% confluence of iPSCs, NutriStem™ (Biological Industries) was refreshed. On day 0, the medium was changed to 3 mL of RPMI (Biological Industries), supplemented with 0.5% L-glutamine (Biological Industries), B27-Insulin (Invitrogen, Carlsbad, CA, USA) and 4.5 µM CHIR-99021 (Tocris, Bristol, UK). On day 2, the medium was changed to 3 mL of RPMI supplemented with 0.5% L-glutamine, B27-insulin, and 5 µM IWP-2 (Tocris). On day 4, the medium was changed to 3 mL of RPMI supplemented with 0.5% L-glutamine and B27-insulin. This medium was refreshed on day 6. On day 8, the medium was changed to 3 mL of RPMI supplemented with 0.5% L-glutamine and B27, and this medium was refreshed on day 10. From day 12, the medium was changed to M-199 (Biological Industries), supplemented with 0.1% penicillin/streptomycin, 5% fetal bovine serum (FBS, Biological Industries), 0.6 mM CuSO_4_·5 H_2_O, 0.5 mM ZnSO_4_·7 H_2_O, and 1.5 mM Vitamin B12 (Sigma-Aldrich, Darmstadt, Germany). This medium was refreshed every other day.

#### 2.5.2. AuNP Implant Preparation

Gd-AuNPs at a concentration of 0.23 M were synthesized as previously described. Then, Gd-AuNPs were encapsulated within 1% *w*/*v* pristine hydrogel to achieve a final hydrogel concentration of 0.6%. For control implants, the hydrogel was diluted with 5% FBS–M199 medium at the same ratio.

#### 2.5.3. Cell Viability

Implant viability was determined using a live/dead fluorescent staining assay with fluorescein diacetate (7 µg·mL^−1^, Sigma-Aldrich) and propidium iodide (5 µg·mL^−1^, Sigma-Aldrich) for 20 min at 37 °C. Live and dead cells within the different implants were visualized by an inverted fluorescence microscope (Nikon ECLIPSE TI-E, Melville, NY, USA), 1 week after the encapsulation.

#### 2.5.4. Implant Immunostaining

Implants were fixed in 4% formaldehyde for 20 min, permeabilized with 0.1% (*v*/*v*) Triton X-100 for 10 min and blocked with 5% bovine serum albumin in PBS for 1 h. Then, implants were stained with primary antibody in 0.5% blocking solution for 2 h, with α-sarcomeric actinin (1:200, ab9465, Abcam, Waltham, MA, USA; Boston, MA, USA). After three washes with 0.5% blocking solution, implants were incubated with Alexa Fluor 647-conjugated goat anti-mouse antibody (1:250; Jackson, West Grove, PA, USA) in 0.5% blocking solution for 2 h. For nucleus detection, implants were incubated for 5 min with Hoechst 33258 (1:100; Sigma-Aldrich) in PBS and washed three times. Samples were imaged using a confocal microscope (Nikon Eclipse NI-E). Images were processed and analyzed using NIS elements software BR 3.2 (Nikon Instruments).

#### 2.5.5. Norepinephrine Response Experiment

iPSCs-derived cardiomyocyte implants were exposed to 1 µM norepinephrine (E4250-1G, SIGMA) solution, and then diluted in HBSS medium for 30 min, in 37 °C. The implants contractions were acquired before and after the exposure to noradrenaline at the same position and magnification, using an inverted fluorescence microscope and ORCA-Flash 4.0 digital complementary metal–oxide semiconductor camera (Hamamatsu Photonics, Hamamatsu city, Japan) at 100 frames·s^−1^. Videos were processed and analyzed using ImageJ software (Java 1.8.0_112).

### 2.6. Phantom Exp

AuNP and Gd-AuNP solutions, the composite hydrogel, pristine hydrogel, and water for control were imaged using a 7T/30 MRI Biospec (Bruker BioSpin, Ettlingen, Germany) equipped with a gradient unit of 660 mT·m^−1^. The following sequence parameters were used: repetition time (TR) = 4.5 ms; echo time (TE) = 2.3 ms; flip angle (FA) = 10°; field of view (FOV) = 25 × 25 mm^2^; matrix size = 192 × 192; slice thickness = 1 mm.

### 2.7. Animal Study

Permission was granted by the Institutional Animal Care and Use Committee (IACUC) of Tel Aviv University, protocol number 04-20-012—“Treatment of biological hydrogel and cardiac cells on cardiac regeneration after ischemia–reperfusion injury in mice model”.

#### 2.7.1. Ischemia–Reperfusion Injury (IRI) Surgery

IRI was induced in C57BL/6 12 week old male mice. Mice were anesthetized with 3% isoflurene, intubated, and ventilated. After the opening of the chest by a left thoracotomy, the major coronary artery was occluded with 0–8 Prolene sutures (W2777, Ethicon) for 45 min. Then, 25 µL of the treatments were injected directly onto the scar tissue (insulin syringe, 30 G, BD Micro Fine Plus). Treatment groups were saline, pristine hydrogel, and composite hydrogels including AuNPs or a 1:1 ratio of AuNPs and Gd-AuNPs. Hydrogels were injected in their solution form at 4 °C, with a final concentration of 0.2%. Then, coronary artery ligation was removed. MRI imaging was performed before the surgery, as well as 1 and 6 weeks post surgery. Mice were sacrificed at the end of the experiment, and their hearts were removed for further analysis.

#### 2.7.2. MRI Imaging

C57BL/6 male mice, weighing from 24 to 32 g, were examined before, as well as 1 and 6 weeks after, the IRI. All experimental procedures were performed under general anesthesia induced by 1–1.5% isoflurane in pure oxygen. MRI imaging was performed using a 7T/30 MRI Biospec (Bruker BioSpin, Ettlingen, Germany) equipped with a gradient unit of 660 mT·m^−1^. MRI scans were acquired using a cross coil setup including an 86 mm resonator and a 20 mm array of surface coil. The body temperature of the animals was maintained by circulating warm water, and all MRI scans were performed with simultaneous ECG and respiration gating. Mice were placed in the magnet with the ears positioned at the isocenter. The MRI protocol included delayed contrast enhancement achieved by intravenous injection of 0.75 mmol·kg^−1^ Gd-DTPA (Magnevist, Bayer HealthCare Pharmaceuticals, Wayne, NJ, USA) following the initial baseline scanning. In order to identify the morphology of the heart tissue, Cine-FLASH (fast low angle shot) MRI was used to image the heart in a short-axis and long-axis view. Six short-axis slices covering the heart from the base to the apex and one long-axis slice were obtained to evaluate myocardial function and infarct size. The following sequence parameters were used: repetition time (TR) = 4.5 ms; echo time (TE) = 2.3 ms; flip angle (FA) = 10°; field of view (FOV) = 25 × 25 mm^2^; matrix size = 192 × 192; slice thickness = 1 mm. Cine movies with 10 frames per slice were reconstructed. Each Cine scan lasted 2 min. In order to identify the scar tissue and the location of Gd-AuNPs within the scar tissue, Cine-FLASH (MRI was used to image the heart in a short-axis and long-axis view. Six short-axis slices covering the heart from the base to the apex and one long-axis slice were obtained to evaluate myocardial function and infarct size. The following sequence parameters were used: repetition time (TR) = 8 ms; echo time (TE) = 3 ms; flip angle (FA) = 25°; field of view (FOV) = 25 × 25 mm^2^; matrix size = 192 × 192; slice thickness = 1 mm. Cine movies with 10 frames per slice were reconstructed. Each CINE scan lasted 2 min.

### 2.8. Assessment of Scar Tissue and Gd-AuNP Location

Image processing analysis was performed on the MRI images. First, the location of Gd-AuNPs was assessed by a segmentation process using the K-means algorithm. In this process, the MRI image was divided into four regions with similar gray values. The AuNP area was labeled in yellow. Next, in order to determine the scar area, image enhancement was improved by image subtraction, where a non-contrast image was subtracted from each post-contrast image. A Despeckle filter was then used to eliminate small areas of noise. The scar area was labeled in red. All measurements were performed in ImageJ, on an end-diastolic volume frame within the same MRI slice.

## 3. Results and Discussion

AuNPs were synthesized as previously described [20]. The particles’ size and shape were assessed using transmission electron microscopy (TEM). TEM micrographs presented spherical AuNPs (Figure 2a) with an average diameter of 30 nm (Figure 2b). The UV/Vis spectrum showed a strong absorption at 520 nm, indicating successful synthesis of the AuNPs (Figure 2c). A useful and common contrast agent for MRI imaging is gadolinium (Gd). Gd can enter the organs and enhance the MRI images by shortening the spin–lattice relaxation time (T1) of the image. As a result, T1-weighted images show brighter signal images. However, the evacuation of Gd after a systemic injection is fast, up to 24 h post injection. Therefore, to allow clear and prolonged imaging of the composite hydrogel by MRI, AuNPs were chemically conjugated with Gd (Figure 2d). FTIR analysis confirmed the presence of Gd on the AuNPs (Figure 2e). The Gd-AuNPs were also investigated for their charge and size according to zeta potential and DLS analyses, respectively. Due to the conjugation of Gd, followed by water absorption, the particle size increased to 181 ± 21 nm (Figure 2f), and the zeta potential increased from −38.5 to −10 ± 3 mV (Figure 2g). The measured size of the Gd-AuNPs in DLS was higher compared to TEM analysis due to the hygroscopic properties of the GdL. TEM images of negative staining of the conjugated particles revealed a gray halo surrounding them (Figure 2h–j). Overall, the results suggested successful modification (Figure 2h–j).

To form the composite hydrogel, Gd-AuNPs were encapsulated within the ECM-based hydrogel to interact with the polymeric backbone. The composite hydrogel was thermoresponsive and formed a gel after heating to 37 °C (Figure 3a). High-resolution scanning electron microscopy (HRSEM) analysis of the composite hydrogel showed the dispersion of Gd-AuNPs on the collagen fibers (Figure 3b,c). Energy-dispersive X-ray spectroscopy (EDX) analysis confirmed the presence of Gd-AuNPs within the hydrogel, as indicated by sharp peaks at 2.1 and 1.2 keV for Au and Gd, respectively (Figure 3d).

The mechanical properties of the nanocomposite hydrogel are extremely important for cell culture and tissue formation [24,25,26,27,28]. Therefore, we next sought to evaluate the effect of NP integration with the hydrogel on the rheological properties. As shown, a significant difference between the mechanical properties of the AuNPs and Gd-AuNPs encapsulated within ECM-based hydrogel could be detected. The nanocomposite hydrogel exhibited higher complex viscosity post Gd coating (Figure 3e,f). This may be attributed to the electrostatic interactions between the Gd complex and the hydrogel fibers.

Next, we tested the biological effect of the Gd-AuNPs on iPSCs-derived cardiomyocytes. The viability of cardiac cells encapsulated within the nanocomposite hydrogel was assessed by a live/dead assay, revealing high viability as within the pristine hydrogel (Figure 4a,b), indicating the biocompatibility of the composite hydrogel. Additionally, the effect of Gd-AuNPs on the morphology and function of engineered cardiac implants was studied. As shown, on days 7 and 14 post encapsulation, the cardiac cells were elongated with massive striation and expressed high levels of sarcomeric actinin (Figure 4c,d). Lastly, the function of the cardiac cells encapsulated within the nanocomposite hydrogel was studied. As shown, the addition of noradrenaline increased the contraction rate of the implants (Figure 4e). Overall, these results indicate that Gd-AuNPs have no negative effect on the encapsulated cells.

We next sought to examine the suitability of the nanocomposite hydrogel for MRI imaging. Commonly, high signal intensity is observed after a Gd injection into scar tissue. However, it was previously found that, at high concentrations (above 0.03 M), Gd results in a hypointense signal [29,30,31]. As the concentration used in our system was 0.23 M, we expected the conjugated Gd to have a dark signal. To confirm this phenomenon, we performed a phantom test. First, we imaged Gd-AuNP and AuNP solutions by MRI. The Gd-AuNP solution showed a hypointense signal compared to the AuNP solution and water (Figure 5a), indicating the conjugation of Gd at high concentration to the AuNPs. Next, we imaged the pristine hydrogel and the composite hydrogel. The MRI scan of the nanocomposite hydrogel revealed black areas that did not appear in the scan of the pristine hydrogel (Figure 5b). We next evaluated the detection of the nanocomposite hydrogel in a dissociated heart. The nanocomposite hydrogel was injected into mouse hearts after ischemia–reperfusion injury. Six weeks later, the hearts were isolated, cut, and used as phantoms. Images support the conclusion that the nanocomposite hydrogel could be accumulated in vivo and detected as a dark signal by MRI (Figure 5d). The signal from the Gd-conjugated AuNP hydrogel was correlated with the injection site (Figure 5c,d). Such a dark signal could not be detected in the MRI scan of untreated hearts (Figure 5e,f). This suggests that, contrary to free Gd that is evacuated from the scar area a few hours post systemic injection, the nanocomposite hydrogel can indicate the injected treatment over time, up to at least 6 weeks, due to the strong bond between the Gd and the AuNPs. Thereby the nanocomposite hydrogel can be used for long-term tracking of the injected treatment.

Next, the Gd-conjugated AuNP nanocomposite hydrogel, nanocomposite hydrogel without Gd coating, pristine hydrogel (without NPS), or saline was intracardially injected into the LV following an IRI. The mice were imaged 1, 4, and 6 weeks post injection by a 7 T animal MRI scan. Figure 6a–d reveal that, in contrast to the saline group (Figure 6a), pristine hydrogel (Figure 6b), and composite hydrogel without Gd coating (Figure 6c), the Gd-AuNP nanocomposite hydrogel (Figure 6d) could be clearly observed in the MRI images 6 weeks post IRI surgery, appearing as a black area within the LV wall (Figure 6a–d). Imaging on weeks 1, 4, and 6 demonstrated that the nanocomposite hydrogel could be detected over a long period of time (Figure 6e,f).

To analyze the location of the composite hydrogel in relation to the scar tissue within the LV wall, image processing was used. For the detection of the composite hydrogel, we used the segmentation process (K-means algorithm). In this process, the MRI image was divided into regions with similar gray values. The area of the nanocomposite hydrogel was easily detected and labeled in yellow (Figure 7a–c). The scar tissue area was detected by subtraction of the same MRI slice before and after systemic Gd-DTPA injection (i.e., free Gd). These changes delineate the areas containing Gd accumulation, which is attributed to scar tissue (labeled in red; Figure 7d–g). Free Gd is evacuated from the healthy myocardium in a few minutes after systemic injection. However, in scar tissue, the Gd cannot be easily evacuated due to the lack of vascularization. Therefore, late gadolinium enhancement (LGE) is an efficient tool for locating and quantifying areas of scar tissue [32,33,34,35,36,37]. This analysis demonstrated the presence of the nanocomposite hydrogel within the scar area (Figure 7h). As the scar tissue and the composite hydrogel appear in different colors, this approach allows noninvasively and effectively detecting scar progress and treatment coverage over time. Furthermore, it allows reinjecting the composite hydrogel to increase its efficacy.

## Figures and Tables

**Figure 1 pharmaceutics-15-01298-f001:**
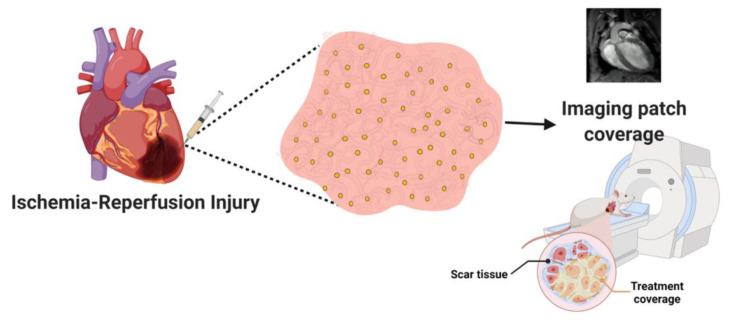
Schematic illustration of the technology. AuNPs are conjugated with Gd and encapsulated within an ECM-based hydrogel. The hydrogel is injected into the left ventricle of mice post ischemia–reperfusion injury, allowing to determine the location of the treatment in relation to scar tissue in the heart.

**Figure 2 pharmaceutics-15-01298-f002:**
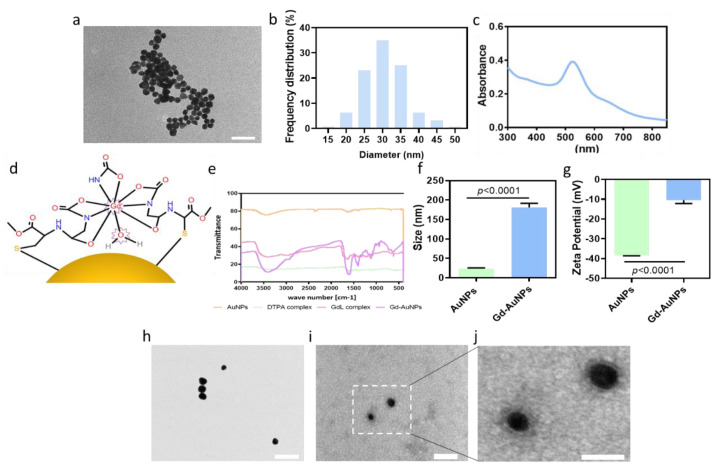
Gd-AuNP characterization. (**a**) TEM image of the pristine AuNPs; scale bar = 200 nm. (**b**) Size distribution of the AuNPs. (**c**) Absorbance spectrum of the AuNPs. (**d**) Scheme of the Gd-AuNPs. (**e**) FTIR analysis of the different steps in Gd-AuNP synthesis: pristine AuNPs in orange, DTPA complex in green, GdL complex in pink, and Gd-AuNPs in purple. (**f**) Diameter of the conjugated AuNPs as indicated by DLS. (**g**) Charge of the conjugated AuNPs as indicated by zeta potential measurements. (**h**–**j**) TEM images of (**h**) pristine AuNPs and (**i**,**j**) Gd-AuNPs after negative staining; scale bars = 100, 50, and 25 nm, respectively.

**Figure 3 pharmaceutics-15-01298-f003:**
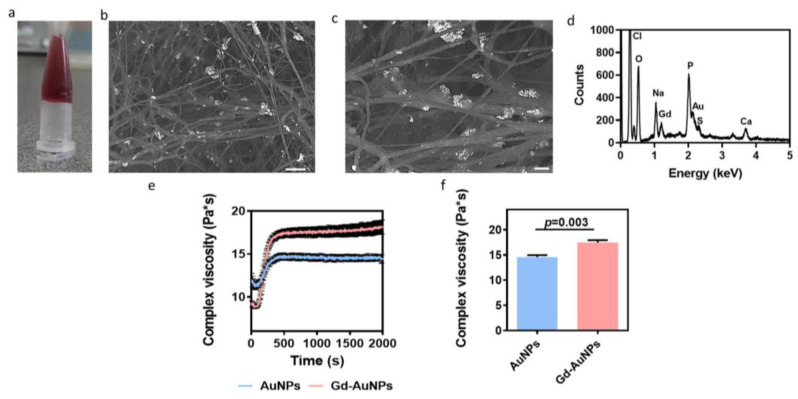
Gd-AuNP nanocomposite hydrogel characterization. (**a**) An image of the nanocomposite hydrogel after gelation at 37 °C. (**b**,**c**) HRSEM images of the nanocomposite hydrogel. The NPs appear in white. Scale bars (**b**) = 1 μm, (**c**) = 300 nm. (**d**) EDX analysis of the composite hydrogel. (**e**,**f**) Rheological properties of the composite hydrogel over time (**e**) and 15 min post gelation in 37 °C (**f**).

**Figure 4 pharmaceutics-15-01298-f004:**
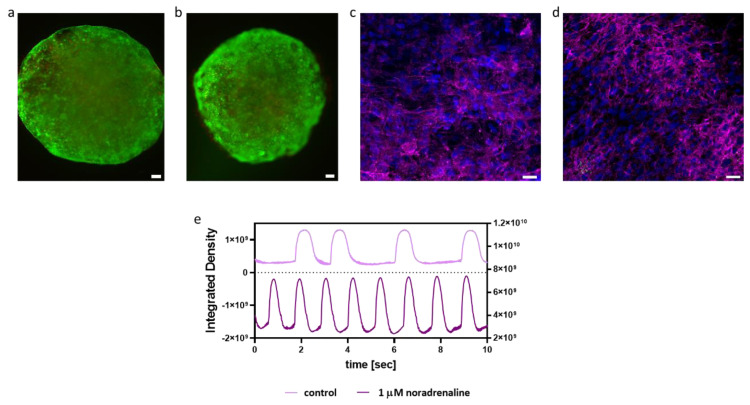
Engineered cardiac implants within the nanocomposite hydrogel. (**a**,**b**) Viability test within the nanocomposite (**a**) and pristine hydrogels (**b**), 1 week post encapsulation; scale bar = 100 μm. (**c**,**d**) Cardiac sarcomeric actinin immunostaining of the nanocomposite implants on day 7 (**c**) and day 14 (**d**). Actinin in pink, nuclei in blue; scale bar = 25 μm. (**e**) Contraction rate of the nanocomposite implant before and after the addition of 1 µM noradrenaline.

**Figure 5 pharmaceutics-15-01298-f005:**
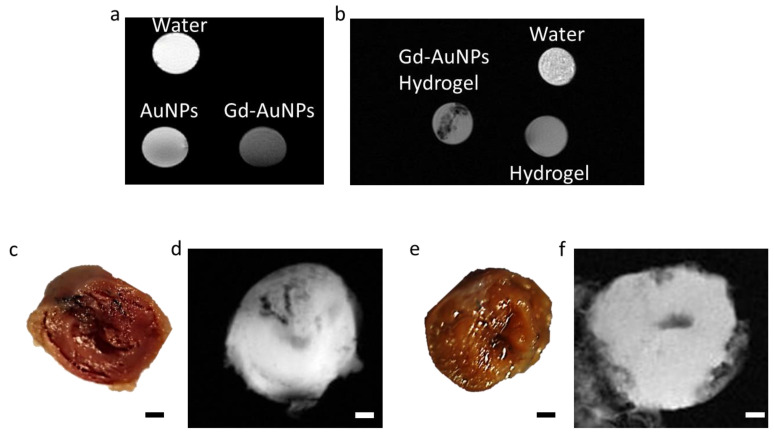
Detection of the nanocomposite hydrogel by MRI. (**a**) MRI imaging of AuNP-Gd, AuNP, and water solutions. (**b**) MRI imaging of droplets of the nanocomposite hydrogel, pristine hydrogel, and water. (**c**–**f**) Images of the heart slice (**c**,**e**) and ex vivo MRI imaging (**d**,**f**) of the treated (**c**,**d**) and untreated (**e**,**f**) hearts, 6 weeks post IRI surgery; scale bar = 1 mm.

**Figure 6 pharmaceutics-15-01298-f006:**
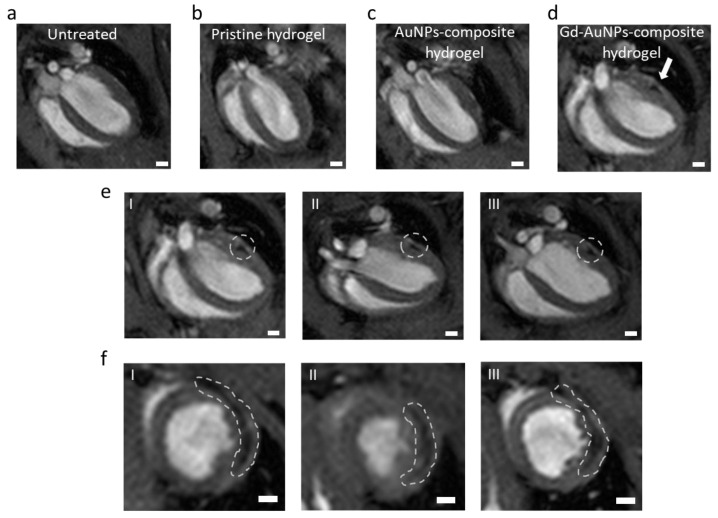
MRI imaging of the heart. (**a**–**d**) MRI images for (**a**) untreated, (**b**) pristine hydrogel-, (**c**) AuNP composite hydrogel-, and (**d**) Gd-AuNP composite hydrogel-treated mice, 6 weeks post IRI. The white arrow in (**d**) indicates the location of the Gd-AuNP composite hydrogel (black area), which was not observed in all other treatments. (**e**,**f**) MRI analysis monitoring the location of Gd-AuNP composite hydrogel (dashed area) over time. (**e**) SAX (short axis slices) and (**f**) LAX (long axis slices) at (**I**) 1 week, (**II**) 4 weeks, and (**III**) 6 weeks post IRI; scale bar = 1 mm.

**Figure 7 pharmaceutics-15-01298-f007:**
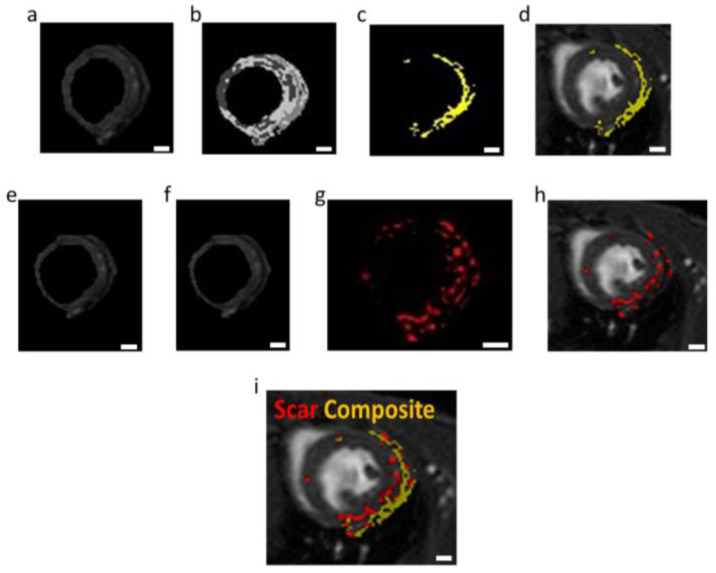
Imaging the coverage area of the treatment in relation to scar area. (**a**–**d**) Segmentation process for detection of the Gd-AuNP nanocomposite hydrogel (yellow). (**a**) LV wall in MRI imaging. (**b**) K-means algorithm, k = 4. (**c**) The location of the Gd-AuNPs composite hydrogel as detected by the algorithm. (**d**) Final MRI image with the location of the hydrogel. (**e**–**h**) Subtraction process for detection of the scar tissue (red). (**e**) LV wall post systemic injection of Gd. (**f**) LV wall before systemic injection of Gd. (**g**) Subtraction MRI image. (**h**) Final MRI image with the location of the scar tissue after noise filtering. (**i**) Merged image of (**d**,**h**). The nanocomposite hydrogel (yellow) with respect to the scar tissue (red), 45 min post systemic injection of Gd-DTPA solution; scale bars = 1 mm.

## Data Availability

Not applicable.

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
