# Peer review of "Imageable AuNP-ECM Hydrogel Tissue Implants for Regenerative Medicine"

_pharmaceutics, 2023, doi:10.3390/pharmaceutics15041298_

Round 1

Reviewer 1 Report

This work presents "Imageable regenerative AuNP-ECM hydrogel tissue implants. This manuscript is recommended to be published after including and addressing the below listed comments with major corrections.

- The authors should eliminate the current grammatical and punctuation mark errors and also confirm the correct scientific English.

- The authors should write the complete terms of all abbreviations (including the instruments) before the first use in the abstract and main manuscript.

- The authors should clearly explain the innovation and importance of their work on the introduction of the manuscript. They should justify the value of the work and compare their work with previously similar published papers. They should develop the advantage and applications of this procedure. The introduction section needs to be elaborated to emphasize the importance of Au nanoparticles and their various applications.

- The authors should cite important references. The below references are suggested to be cited in the revised manuscript:

Inorganic Chemistry Communications 130, 108746 (2021)

Author Response

This work presents "Imageable regenerative AuNP-ECM hydrogel tissue implants. This manuscript is recommended to be published after including and addressing the below listed comments with major corrections.

  1. The authors should eliminate the current grammatical and punctuation mark errors and also confirm the correct scientific English.

Thank you for this comment. The manuscript has been revised.

  1. The authors should write the complete terms of all abbreviations (including the instruments) before the first use in the abstract and main manuscript.

Thanks for the important comment. We revised the text accordingly.

  1. The authors should clearly explain the innovation and importance of their work on the introduction of the manuscript. They should justify the value of the work and compare their work with previously similar published papers. They should develop the advantage and applications of this procedure. The introduction section needs to be elaborated to emphasize the importance of Au nanoparticles and their various applications.

Thanks for this comment. A short description of the innovation and the advantages of our work was added to the introduction. According to our knowledge, there is no any similar study that images the injectable treatment in relation to the actual scar tissue in MRI.

  1. The authors should cite important references. The below references are suggested to be cited in the revised manuscript:

Inorganic Chemistry Communications 130, 108746 (2021)

Thank you for this comment. We added several references to the manuscript.

Reviewer 2 Report

The authors developed an injectable hydrogel that can support the cardiac cell growth and promote cardiac tissue assembly. In addition, it can cover the scar area and show imaging function by MRI. The idea was good and the development of the hydrogel may improve the accuracy of a tissue engineering treatment. However, there are some problems with the result presentation and some of the important points are missing. Therefore, I suggest the acceptance after revising the following points.

1.     In figure 2, the authors provided the DLS analysis of Gd-AuNPs, which is 181 nm. However, in figure 2i-j, the TEM results indicated the size of Gd-AuNPs is less than 30 nm, which is not consistent to the DLS results.

2.     The arrange of results should be in a logical order. For example, in line 278-279, “The nanocomposite hydrogel exhibited higher complex viscosity post Gd coating (Fig. 3f and e)”, the better order should be (Fig. 3e and f). Also, in figure 3f, the AuNPs should be in the left and the Gd-AuNPs should be in the right, which follows the logic of your manuscript as previous.  

3.     In figure 3e, the thickness of lines should be adjusted so that the format of the figures can be unified.

4.     The scale bars of figure 5c-f, 6 and 7 should be provided.

Author Response

The authors developed an injectable hydrogel that can support the cardiac cell growth and promote cardiac tissue assembly. In addition, it can cover the scar area and show imaging function by MRI. The idea was good and the development of the hydrogel may improve the accuracy of a tissue engineering treatment. However, there are some problems with the result presentation and some of the important points are missing. Therefore, I suggest the acceptance after revising the following points.

  1. In figure 2, the authors provided the DLS analysis of Gd-AuNPs, which is 181 nm. However, in figure 2i-j, the TEM results indicated the size of Gd-AuNPs is less than 30 nm, which is not consistent to the DLS results.

Thank you for the important comment. Due to the hygroscopic properties of the GdL, Gd-AuNPs absorbed water molecules during the DLS measurements; therefore, the size was larger compared to the TEM analysis. We added this explanation to the manuscript. The text now reads: " The measured size of the Gd-AuNPs in DLS is larger compared to TEM analysis…"

  1. The arrange of results should be in a logical order. For example, in line 278-279, “The nanocomposite hydrogel exhibited higher complex viscosity post Gd coating (Fig. 3f and e)”, the better order should be (Fig. 3e and f). Also, in figure 3f, the AuNPs should be in the left and the Gd-AuNPs should be in the right, which follows the logic of your manuscript as previous.

Thanks for this comment. We revised the results accordingly.

  1. In figure 3e, the thickness of lines should be adjusted so that the format of the figures can be unified.

Thank you for this comment.  

  1. The scale bars of figure 5c-f, 6 and 7 should be provided.

Thank you for this important comment. The scale bars were added to the figures.

Round 2

Reviewer 1 Report

The manuscript can be accepted for publication.